# Identifying research priorities for patient safety in mental health: an international expert Delphi study

Lindsay H Dewa,[1,2] Kevin Murray,[3] Bethan Thibaut,[2] Sonny Christian Ramtale,[1] Sheila Adam,[1] Ara Darzi,[1] Stephanie Archer[1]

## ABSTRACT

**Objective** Physical healthcare has dominated the patient safety field; research in mental healthcare is not as extensive but findings from physical healthcare cannot be applied to mental healthcare because it delivers specialised care that faces unique challenges. Therefore, a clearer focus and recognition of patient safety in mental health as a distinct research area is still needed. The study aim is to identify future research priorities in the field of patient safety in mental health.

**Design** Semistructured interviews were conducted with the experts to ascertain their views on research priorities in patient safety in mental health. A three-round online Delphi study was used to ascertain consensus on 117 research priority statements.

**Setting and participants** Academic and service user experts from the USA, UK, Switzerland, Netherlands, Ireland, Denmark, Finland, Germany, Sweden, Australia, New Zealand and Singapore were included.

**Main outcome measures** Agreement in research priorities on a five-point scale.

**Results** Seventy-nine statements achieved consensus (>70%). Three out of the top six research priorities were patient driven; experts agreed that understanding the patient perspective on safety planning, on self-harm and on medication was important.

**Conclusions** This is the first international Delphi study to identify research priorities in safety in the mental field as determined by expert academic and service user perspectives. A reasonable consensus was obtained from international perspectives on future research priorities in patient safety in mental health; however, the patient perspective on their mental healthcare is a priority. The research agenda for patient safety in mental health identified here should be informed by patient safety science more broadly and used to further establish this area as a priority in its own right. The safety of mental health patients must have parity with that of physical health patients to achieve this.

[1]NIHR Imperial Patient Safety Translational Research Centre, Imperial College London, London, UK
[2]School of Public Health, Imperial College London, London, UK
[3]Forensic Mental Healthcare, West London Mental Health NHS Trust, London, UK

**Correspondence to**
Dr Lindsay H Dewa;
l.dewa@imperial.ac.uk

## Strengths and limitations of this study

► Service user experts were involved in both the semistructured interviews and survey stages.
► The survey response rate was good overall and increased over each Delphi round.
► The selection of experts were mainly from the USA and UK, particularly the service users, therefore priorities may be generalisable to only a limited extent.
► The field of patient safety in mental health is relatively new, and as such, experts may be more likely to identify themselves as experts in mental health rather than patient safety; experts may therefore have been missed.

faces particular challenges.[1 2] For example, some patients receiving mental healthcare may lack mental capacity and may have their freedom limited to a greater or lesser extent. In addition, there is longstanding evidence that mental health services remain under-resourced at a time when need appears to be increasing.[3] Mental health is a relatively neglected area, especially in acknowledging the patient voice in relation to care and safety.[4] Furthermore, stigma and discrimination surrounding mental health issues has the potential to contribute to patient safety being neglected.[1]

'Patient safety' in mental health is difficult to define because of the often interrelated understanding of 'disorder' and 'behaviours' for mental health patients within the provision of healthcare. For example, suicide and self-harm are unsafe behaviours not diagnoses. Therefore patient safety in relation to mental health should mean the avoidance of unintended unsafe or iatrogenic harm associated with mental healthcare (either an error in inappropriate treatment or an omission to detect unsafe behaviour).

Some types of adverse events that occur due to failure in patient safety in mental

## INTRODUCTION

Patient safety within mental healthcare has not been researched to the same extent as patient safety within physical healthcare.[1] The applicability of findings from physical health cannot simply be applied to mental health because it is a specialised area of care that

healthcare are similar to those in physical healthcare, for example, medication errors, misdiagnosis and accidents. In contrast, unsafe patient behaviours, such as absconding, violence, self-harm and suicide constitute poor outcomes specific to mental healthcare.[2] These may arise from factors inherent in the disorders themselves, giving rise to increased risk to the patient, other patients and to staff and features of the physical environment and the demanding work environment for staff.[1]

Consequently, unsafe patient behaviours within mental healthcare require specialised management strategies distinct to those in physical healthcare. They may require coercive management strategies such as observation, locked wards, seclusion, restraint and enforced medication, which relinquish patients' autonomy. While there has been some attempt to reduce coercion in mental health settings; replaced with de-escalation techniques (eg, Safewards, active communication),[5] there is still a need to reduce the instances of coercive control further, by staff sharing control with patients over their own care. Indeed, active strategies are in place to help prevent these adverse incidents, with a specific focus on supporting research[6] but a clearer focus and recognition of patient safety in mental health as a distinct research area is still needed.

Despite the wide range of adverse events, research pertaining to patient safety and mental healthcare is limited. Only three papers have examined priorities in patient safety in a mental health context; they have used a range of research methods including the Delphi technique.[1 7 8] Brickell and colleagues conducted a literature review,[1] key informant qualitative interviews and a round-table event with selected Canadian experts, with the latter published separately.[2] Making safety culture a priority, standardisation of terminology, practices and policies across Canadian mental healthcare were deemed the most important recommendations. Mascherek and Schwappach identified priorities in patient safety in mental healthcare for Switzerland using expert panels, round-table discussion and a modified Delphi technique.[7] There were nine priority topics but these generally covered iatrogenic safety incidents including diagnostic errors, communication errors and management errors (eg, aggression management against self and others, unnecessary use of coercive measures). Cowman and colleagues recently conducted a large Delphi study with 2809 staff across 17 countries in Europe to identify current management practices and future priorities relating to violence management in mental health services.[8] Key future priorities in violence management included aetiology, prevention, environmental influence and best management practice.

There are limitations in each of the three studies, and none looked at all aspects of patient safety in mental health internationally. First, two studies focused on priorities in their own countries only, and the results cannot be extrapolated to the wider international platform.[1 7] Second in one paper, sample size was low (round 1: n=11;

round 2: n=14) and attrition rate high across Delphi rounds.[7] Third, two studies only alluded, to research priorities, and this was either not part of the research aim[1] or in the context of violence management only.[8] Finally, and perhaps most importantly, the level of patient involvement, in defining research priorities is unclear.

The aim of the current study is to use expert consensus building methodology[9] with a group of international academic and service user experts to identify the future research agenda in this area. We refer to 'service user experts' when we are discussing the Delphi participants and the statement generating interviews; we refer to 'patients' when we are including the language of the specific statement evaluated.

## METHODS

### Design

A Delphi technique was used to build consensus on research priorities for patient safety in mental health (see figure 1). The Delphi method is an iterative a priori process in which a group of expert stakeholders come to a structured consensus view on a particular topic through a number of rounds with controlled feedback.[10] Delphi studies have been successfully conducted to establish research priorities for numerous different topic areas including prison health, mental health and occupational mental health.[11–13]

### Expert identification and semistructured interviews

Potential academic experts were identified through a systematic review currently in progress[14] in addition to hand searching of articles and word of mouth. Academic experts satisfied the following inclusion criteria: published at least six articles in patient safety in the mental health field; had at least 5 years experience in patient safety in mental health and established reputation in patient safety in mental health, defined as having a high number of citations, a role at a national level or having made a significant impact to the academic field. A provisional list of 15 international academic experts was identified; individuals were contacted via email in the first instance and followed up via phone after a week if there was no response. Participating academic experts were then encouraged to identify other experts through a snowballing technique.

Service user experts were recruited through a UK-based independent third-sector group. Inclusion criteria included: personal experience of mental health services and an ability to comment on patient safety-related issues. Service users were excluded if they were unable to provide informed consent due to being too physically or mentally unwell or if they were non-English speaking.

Over a 2-month period (February to March 2017), semi-structured telephone interviews were conducted with the academic experts to ascertain their views on research priorities in patient safety in mental health. Face-to-face interviews were conducted with service user experts at a place convenient to them. All interviews were recorded. As

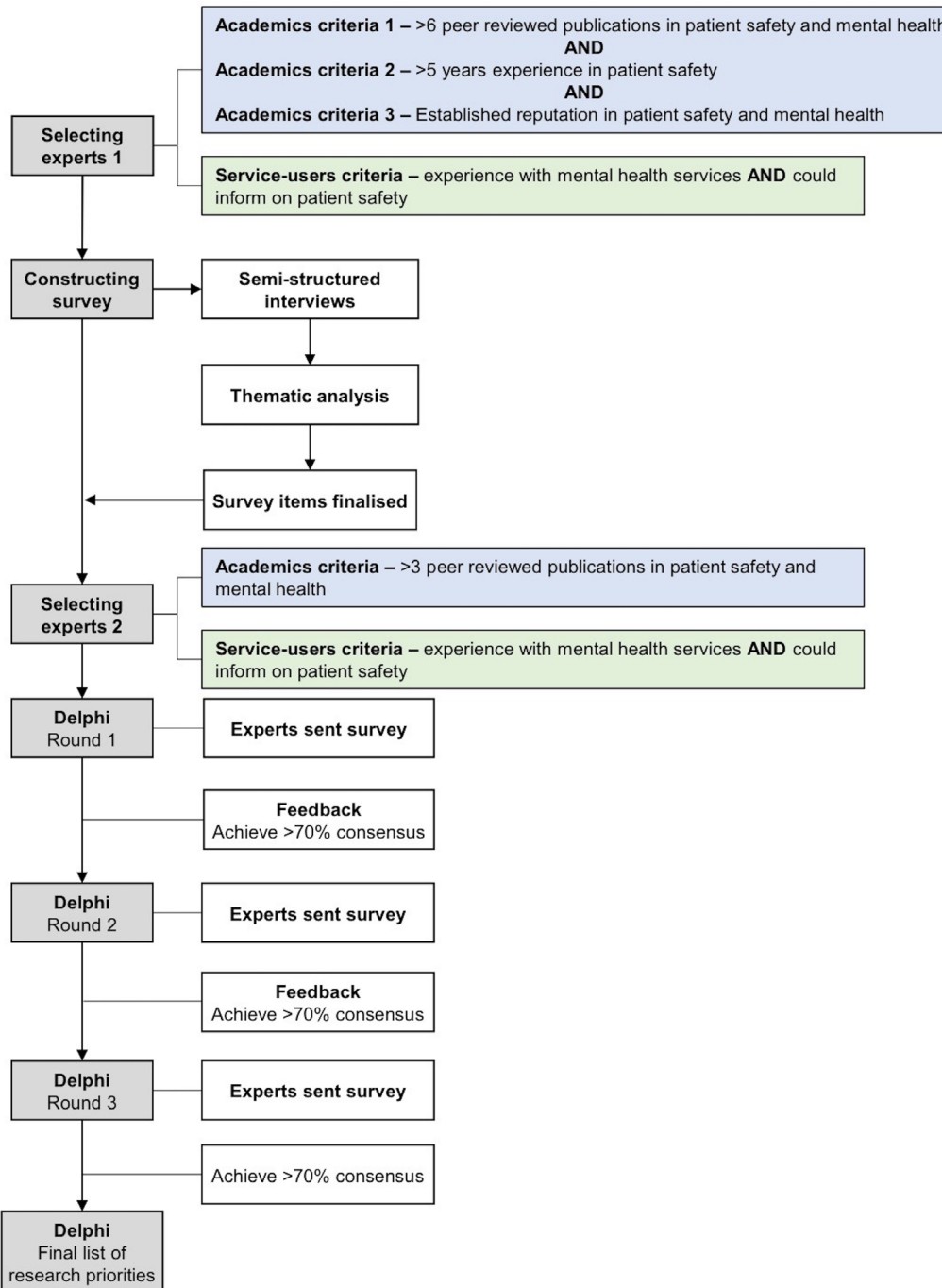

**Figure 1** Delphi diagram of study process.

concrete dimensions of patient safety in mental health do not exist, the topic guide was intentionally broad to bring experts' own knowledge, experience and understanding to the consensus. However, it did cover general questions about known patient safety incidents (eg, suicide, self-harm, etc) and probing questions on existing patient safety management (eg, suicide prevention strategies). The topic guide was reviewed by the research team and structured to focus on three main areas: (1) experience of patient safety in mental health; (2) current research in the area; (3) research gaps and future research priorities. All audio files from the interviews were transcribed verbatim.

Themes were extracted from the transcripts, checked by S Archer and were used to construct a survey containing statements pertaining to each priority identified.

### Delphi round 1

In April 2017, the survey was circulated electronically to a group of international experts. This group comprised the interviewed academic and service user experts, other experts they recommended and other selected academic experts who had published more than or equal to three peer-reviewed papers[9] in the field of patient safety in mental health. Potential service user experts

were approached by email by selected third-sector organisations. The inclusion criteria for service users was the same as the interview criteria. The survey was also advertised on Twitter and in a third-sector newsletter. When completing the survey, experts were asked to indicate anonymously how much they agreed with each statement. Agreement was measured on a five-point Likert scale: strongly agree, agree, neither agree or disagree, disagree and strongly disagree. The Delphi process concluded when either consensus relating to strongly agree or agree for every individual statement was obtained (>70%) or three rounds were completed in line with guidance.[15 16] As consensus (>70%) was not achieved for all statements in round 1, a further Delphi round was pursued to establish if any additional priorities would achieve consensus.

### Delphi round 2

Group feedback and a summary of the collated statement scores that did not reach agreement was then circulated to the experts. Experts were given a further 2 weeks to repeat the survey, choosing whether to amend their scores based on the summary information provided or keep their original score. Once submitted, the new scores were summarised by the researchers and assessed for consensus across the expert group. Consensus was not obtained for all statements, so a third round was pursued to establish any additional research priorities.

### Delphi round 3

The distribution and analysis of scores were repeated as in round 2. Experts were given a further 2 weeks to repeat the survey. We satisfied round 3 criteria and the Delphi rounds ceased.

### RESULTS

#### Semistructured interviews

Nine academic experts participated in the semistructured interviews (female; n=3). Four experts were based in England, two in the USA and the remaining experts were in Australia, Bahrain and Scotland. The mean number years of experience in the field of patient safety in mental health was 23.22 (SD 8.20) and ranged from 10 to 36 years. Participants had at least 10 publications each in the area of patient safety in mental health (table 1). Four female service user experts were also interviewed who fulfilled the inclusion criteria (table 2).

In general, both academic and service user experts believed the patient perspective was needed to further understand patient safety in mental health field; however, some academic experts felt this might be difficult in practice, especially with patients with more severe mental health symptoms. Experts agreed that better understanding of physical health adverse events in mental health patients was a research priority. Academic experts generally agreed more research looking at suicide prevention was needed, but service users spent limited time discussing this. Instead, service user experts explored

**Table 1** Academic experts from semistructured interviews* (n=9)

| Academic | Gender | Academic role | Country | Publications | Experience in the field (years)† | Preventing suicide | Management of self-harm | Reducing coercive practice | General patient safety in mental health |
|---|---|---|---|---|---|---|---|---|---|
| Academic 1 | Female | Associate professor of psychiatry and behavioural sciences | USA | 25 | 36 | X | | | |
| Academic 2 | Male | Professor of psychiatry | England | 100+ | 25 | X | X | | |
| Academic 3 | Male | Adjunct associate professor of psychiatry | USA | 21 | 17 | X | | | |
| Academic 4 | Male | Senior lecturer | England | 10 | 30 | X | X | | |
| Academic 5 | Male | Deputy medical director | Scotland | 10+ | 10 | | | | X |
| Academic 6 | Male | Professor of nursing | Bahrain/Ireland | 15+ | 20 | | | | X |
| Academic 7 | Male | Professor of mental health | England | 10+ | 30 | | | | X |
| Academic 8 | Female | Associate professor of mental health nursing | Australia | 15 | 16 | | | | X |
| Academic 9 | Female | Professor of mental health nursing | England | 30+ | 25+ | | | | X |

Patient safety in mental health academic expertise — Established reputation topic area

*No interests declared.
†At time of publication.

| Table 2 | Service user experts from semistructured interviews (n=4) | | | | |
|---|---|---|---|---|---|
| Service user | Gender | Country | Years since last access to mental health services | Type of experience |
| Service user 1 | Female | England | >1 | Adult mental health services |
| Service user 2 | Female | England | <1 | Adult mental health services |
| Service user 3 | Female | England | >1 | Adult mental health services |
| Service user 4 | Female | England | >1 | Adult mental health services |

opportunities for research that were based on their own experience of being a psychiatric inpatient. They spoke about investigating possible alternatives to, and factors directly related to, the experience of coercive intervention in addition to alternatives in being admitted to hospital. One hundred seventeen priority statements were subsequently extracted from the semistructured interviews (see online supplementary files 1 and 2).

### Delphi survey

Forty-two participants took part in the Delphi survey. Two-thirds were academic experts (n=28; 66.6%) of which 42.9% were female (n=12). Half of the academic experts were based in either the UK or the USA (n=14; 50.0%). The remaining experts were in Switzerland, Netherlands, Ireland, Denmark, Finland, Germany, Sweden, Australia, New Zealand and Singapore. The mean years of academic experience in the patient safety in mental health field was 4.8 (SD 57). Main areas of expertise covered aggression, violence and coercion, violence and suicide prevention; the use of and reduction of restrictive practice including restraint and seclusion and risk management.

One-third of participants were service user experts (33.3%; n=14) from the UK. Two-thirds were female (64.3%; n=9) and most had experience with adult mental health services (92.9%; n=13).

In round 1, 38/117 statements reached consensus (>70%). The two expert groups (service users and academics) showed modest agreement in their responses (r=0.25 p<0.01). The 79 statements that did not achieve consensus were presented to participants again in round 2. Three quarters of participants participated in round 2 (76.2%; n=32). An additional 35 statements reached consensus. Two participants dropped out before round 3. Subsequently, 30 participants were contacted to take part in round 3. In this final round, 28 participants completed the survey (87.5%). A further six statements reached consensus (>70%). In total, 79 statements achieved consensus.

## Main areas of agreement

### Patient perspective and safety planning

Understanding how patients can contribute to their own safety emerged as a high research priority. Patient perspectives on medication safety, safety culture in those who self-harmed and the ways staff could help manage their self-harm were also considered a high priority (90.6%, 90.6% and 73.8%, respectively). Experts had agreement on patient-centred research priorities including what constitutes good self-driven safety planning (90.6%), understanding patient's perception of their own risk factors (71.4%) and an exploration into a personalised model of risk (81.3%). Consensus was also obtained on research exploring the patient perspective on staff violence on patients and ligature points (both 78.1%).

### Physical health in mentally ill patients, mental health and physical health comparison and patient safety in the community

Experts indicated that exploring physical health adverse events in mental health patients should be a priority (81.3%). To a lesser extent, staff engagement with mental health patients about their physical health (75.0%) and comparison of physical and mental health hospitals on patient safety and errors (71.9% and 71.4%, respectively) were seen as research priorities. Consensus was also achieved in a research priority exploring the relationship between patients waiting to be seen in accident and emergency rooms and the link to general violence (78.1%).

### Suicide prevention

Experts agreed that studies looking at suicide prevention interventions in large samples, marginal groups and outpatients were research priorities in the patient safety in mental health field (83.3%, 81.0% and 81.0%, respectively). There was also a preference for research exploring suicide prevention in community samples. For example, the detection of people who are at risk of suicide but not in the mental health system; risk factors for suicide on discharge from hospital and a comparison between the community management, general hospital and psychiatric hospitals on their approach for suicide prevention were deemed important future research areas (78.6%, 78.1% and 71.4%, respectively). A need to look at adverse events other than suicide in mentally ill patients was also evident (81.0%)

### Safety culture and physical environment

There was clear preference for research on making psychiatric inpatient settings safer for patients. Priorities included the best approach to ensuring a safe environment in mental health units (85.7%), understanding and improving safety culture (83.3%) and identification of environmental factors that indicate a safe environment (81.0%). Likewise, experts wanted to know what environmental factors influence violent incidents and to observe adverse events on the wards (76.2% and 85.7%, respectively). Explicit research priorities within this safety culture and environment section included an exploration of the relationship between therapeutic engagement and

patient safety, general preventable errors and the safe alternatives to admitting someone to hospital, all of which gained a high-level consensus (81.0%, 78.2% and 81.0%, respectively). The establishment of best practice guidelines for the design of inpatient settings was also strongly supported (87.5%).

### Restraints

Experts agreed that research into the contributory factors to restraint, specifically in patients with physical health problems was a priority (85.7%). The prevalence and reasons for differences in restraint across units were also a priority (71.4%). Two priorities focused on understanding the individual perspective in relation to restraint, including exploration of patient trauma after restraint (78.6%) and how people feel about restraint in general (71.9%). Finally, experts agreed that research was needed to examine factors allowing for reduction in restrictive practice (90.5%) and alternatives to restraint (88.1%).

### General safety, risk and violence management

Two research priorities that aimed to examine the nature and benefits of risk assessment (75.0% and 78.1%, respectively) were supported by experts. Additionally, experts indicated positive and negative factors that influence violence management and more protective factors for violence in mental health patients were deemed necessary future research areas (both 78.6%). Furthermore, experts agreed that the role of racism in relation to coercive interventions should be examined (71.4%). Other proposed priorities with strong agreement included staff decisions to admit patients presenting with self-harm behaviour (87.5%), what de-escalation should look like and how to evaluate it (87.5%) and exploration of staff attitudes on coercive practice (75.0%).

### Psychological trauma

The expert group agreed that the mechanism of trauma and psychological harm associated with inpatient admission was a research priority (78.1%). There was also consensus that the patient experience of coercion in those who had experienced trauma should be examined (78.1%).

### Children's safety

Experts indicated that research exploring the impact of adverse events for parents on the psychological well-being of their children was an important future research area. An understanding of predictive factors in self-harm and suicide in children and an exploration of a child's safety when parents with mental health problems become unwell both achieved consensus (83.3% and 88.1%, respectively).

## DISCUSSION
### Main results and comparison to other studies

This is the first international Delphi study exploring research priorities in patient safety in mental health from both the academic and service user perspective.

Sixty-eight per cent of statements were identified as future research priorities in patient safety in mental health (n=79). The top six research priority areas are shown in box 1; half of these areas were identified by patients for consideration. This suggests a need to shift the thinking about patient safety in mental health services to that of acknowledging the patient voice in relation to their own care and safety. This is in line with other clinical domains that have had success with patient involvement in achieving safer services. For example, some interventions that centre on error prevention through promotion of complex behavioural change and patient involvement have been deemed effective.

Thirty-two per cent of statements were not seen as priorities. Two research priority statements that concerned research identifying specific places of safety in the community had borderline consensus >65%; but in general, patient safety in community mental health settings were not seen to be a research priority. This is surprising given that most mental healthcare takes place in the community. Also surprising, considering that this was an international study that included respondents from the USA, was the finding that participants disagreed/strongly disagreed that access to guns in the community after discharge and omissions in prescriptions were research priorities (37.5% and 34.4%). This maybe because the majority of survey experts were from countries where gun violence may not be prevalent in the general community, including among mental health patients.

Research on medication safety is seen as a priority in physical healthcare safety.[17] As such, we expected medication safety to also be identified as a research priority in mental healthcare; however, this was not the case. Although 'patient perspectives on medication safety' had strong consensus (box 1), the remaining statements related to medication safety were not considered research priorities. Medication safety was not seen as a research priority overall; however, this may have been because the statements focused on minority groups including the elderly and patients with a dual diagnosis, and therefore the statements may not have been applicable to most experts, especially service users.

In general, statements that proposed longitudinal trials and research focusing on Safewards did not achieve consensus. It is possible that as longitudinal studies are deemed expensive and have added temporal demands,[18]

---

**Box 1 Top six future research priorities that achieved the highest consensus**

- ► Patient contributions to their own safety.
- ► The patient perspective on medication safety.
- ► Perspectives on safety culture in patients who self-harm.
- ► Good self-driven individualised safety planning.
- ► Safety plans and safety improvement.
- ► Factors in allowing reduction in restrictive practice including restraint and seclusion.

academic experts felt they would require a large programme grant and therefore were unattainable as a timely research priority. It is also possible experts felt de-escalation techniques may have had some focus in recent years,[5 19] and therefore other areas required more focus going forward. Specifically, the identification of positive, protective and negative factors for violence were still deemed necessary research areas.

Some areas were not endorsed as priorities by the semistructured interviews, including failure to establish diagnosis, the correct diagnosis and to deliver appropriate treatment. In addition, delays in accessing appropriate care because a bed is not available or service is unavailable were also not mentioned. This is surprising because it has been considered an important barrier to improving mental healthcare service provision[20]; however, it is possible that experts considered this a patient care or quality improvement issue and not patient safety.

### Strengths and limitations

This is the first Delphi study to collate research priorities across all aspects of patient safety in mental health internationally, over four continents. A survey sample, which was notably larger than another Delphi study in the patient safety mental health field,[7] was obtained. A key strength is the involvement of service user experts in both the semistructured interviews and survey stages. Another key strength is that all but one interviewed academic experts took part in the survey. The overall survey response rate was good and increased rather than decreased over each round (round 2, 76.2%; round 3, 87.5%).

However, there are some methodological limitations. First, experts were mainly from Europe, the USA and other high-income countries; therefore, the established set of research priorities may only reflect a research agenda for developed countries. Second, only English-speaking women service users participated in the interviews and therefore priorities may not be generalisable to the wider service user population. Third, it was not easy to define an academic expert in the patient safety and mental health field; unlike another expert consensus building study.[9] This was because the patient safety in mental health field is still relatively new with experts likely to identify themselves as experts in mental health rather than patient safety. It is therefore possible academic experts were missed, unlike in another Delphi study.[8] Fourth, experts who took part in the survey were not asked about their awareness of existing research in the area. Therefore, experts, particularly service users, could have recommended research that had already been covered. Furthermore, the survey did not offer an opportunity to suggest additional areas for consideration that were not identified from the interviews. Finally, participant fatigue was a possible contributory factor towards latter statement completion, particularly due to the large amount of statements (n=117).

### Implications

This Delphi study should inform the research agenda for patient safety in mental health field, specifically in the UK, USA and other developed countries. It is recommended that academics wishing to pursue research in these areas conduct a comprehensive literature review pertaining to each priority area as a first step. Subsequent studies representing each priority area are then needed to target specific research areas, study types and methodologies. The research findings from these studies should be included in future policy and practice.

This study has included the service user's 'voice' when identifying research priorities in patient safety in mental health field. Following this, it would be useful to cocreate future research with service users, their families and carers, psychiatric nurses and other mental healthcare professionals with the aim of informing studies with an expert perspective. This is in line with current guidance and the increased expectation to involve a range of experts, including service user experts in the design, production and dissemination of research.[21]

## CONCLUSION

This is the first international expert consensus study to identify research priorities in patient safety in the mental health field. Experts identified what they considered to be the most important areas for future research; these include the patient perspective on their mental healthcare, including medication safety, safety planning and self-harm management. While this Delphi study shows consensus in several areas, not all of these priorities are firmly on the research agenda for patient safety in mental health.

To establish patient safety in mental health as a research priority in its own right, prospective academic research programmes with appropriate funding are required. This will encourage research that helps detect deterioration of mental health, develops safety interventions and generally improves safety. While mental healthcare provision is increasingly seen as a priority, there is still work to be done; the safety of mental health patients must have parity with patient safety within the physical health field.

**Acknowledgements** The authors thank Dr Martine Nurek for her analytical support and all experts for their participation in this study.

**Contributors** LHD drafted and finalised the protocol, designed the study, led the data collection and analysis and interpreted the data. LHD drafted the manuscript and is also the corresponding author. KM advised on clinical definitions, mental health content interpretation and intellectual content. KM, BT, SCR and S Archer were involved in drafting the manuscript and approved the final version to be published. BT contributed to data collection. S Adam advised on clinical definitions, critically reviewed intellectual content and was involved in the drafting and critical appraisal of the manuscript. She also approved the final manuscript. AD approved the final manuscript to be published. S Archer contributed to designing the protocol, advised on service user recruitment and critically reviewed intellectual content. She was also involved in critical appraisal of the manuscript and approved the final version.

**Funding** This work was supported by the National Institute for Health Research (NIHR) Patient Safety Translational Research Centre via an NIHR programme grant.

**Competing interests**  None declared.

**Patient consent**  Not required.

**Ethics approval**  The study was approved by the Imperial College Research Ethics Committee.

**Provenance and peer review**  Not commissioned; externally peer reviewed.

**Data sharing statement**  No additional data are available.

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
