## [Reviewer comments · BMJ Open]

ARTICLE DETAILS

TITLE (PROVISIONAL)	Identifying research priorities for patient safety in mental health: an international expert Delphi study
AUTHORS	Dewa, Lindsay; Murray, Kevin; Thibaut, Bethan; Ramtale, Sonny; Adam, Sheila; Darzi, Ara; Archer, Stephanie

VERSION 1 – REVIEW

REVIEWER	Anthony Jorm University of Melbourne, Australia
REVIEW RETURNED	10-Jan-2018

GENERAL COMMENTS	Identifying research priorities for patient safety in mental health: an international expert Delphi study Anthony Jorm University of Melbourne, Australia 10-Jan-2018 This paper reports a well-conducted Delphi study. I have the following suggestions for improvement: 1. Any consensus criterion is arbitrary, but did the authors have any reason for choosing 70% as the cutoff for consensus?2. On page 10, give the number of academic experts who participated in the Delphi study. Currently, only the percentage is given.3. It is common in Delphi studies to give panel members feedback on responses from previous rounds. Was any feedback given in this study?4. Can the authors provide any quantification of agreement between academic and service user experts? The methodological article by Jorm on the Delphi method in mental health research (Aust N Z J Psychiatry. 2015 Oct;49(10):887-97) reports a number of studies that have looked at agreement among professional, consumer and carer experts. It would be informative to have such data from this study.5. When reporting the results, the authors repeatedly refer to consensus across both academic and service user experts. However, the two groups were treated as one panel and only 14 out of 42 were service users. It would therefore be possible for an item to get 70% endorsement if all the academics endorsed it but only a minority of service users. I think it would only be appropriate to draw conclusions about endorsement by both groups if their consensus was assessed separately.
---

	6. On page 12, the authors state that “The majority of experts, both academic and service users agreed...”. If the consensus criterion has been set at 70%, the authors should stick to this and not use a majority (i.e. >50%) as a criterion. 7. I think a major limitation, which the authors don’t mention, is that experts may not know what research has already been done. They were not provided with any information about the current state of research and may be suggesting areas as a priority which are already well covered. In particular, service users may not know what the existing research is, and even academic experts may not be fully aware of what is happening outside of their immediate area of expertise. 8. Another limitation is that the Delphi questionnaire was based on the interviews with 9 academics and 4 service users. There was no opportunity for the Delphi panelists to suggest additional areas for consideration in the questionnaire that were not in the interview content. 9. On page 16, I was unclear what “attribution bias” was in this context. 10. The Supplementary File gives all items, including both those that reached consensus and those that did not. I would find it easier to read the findings if those that reached consensus were presented separately from those that did not.
--	---

REVIEWER	Miranda Wolprt UCI, UK
REVIEW RETURNED	25-Jan-2018

GENERAL COMMENTS	I thought this was a clear and interesting paper and will be helpful to take forward debate and further research in this area for example with a specific focus on community provision and particular groups e.g. children and young people. My only minor comment is that it would be helpful to clarify the key finding in the abstract which is currently so truncated as to be a bit unclear as to what the consensus actually was re top research priorities.
--

VERSION 1 – AUTHOR RESPONSE

Dear Dr Gray,

Re: Submission of revised paper: Identifying research priorities for patient safety in mental health: an international expert Delphi study (ref: bmjopen-2017-021361).

Thank you for your email dated 29/01/2018 enclosing the reviewer’s comments. We have carefully reviewed the comments and have revised the manuscript accordingly. Our responses are given in a point-by-point manner below. Changes to the manuscript are showed below.

Point by point response

Reviewer 1

1. Any consensus criterion is arbitrary, but did the authors have any reason for choosing 70% as the cutoff for consensus?

We thank you for your critique. An explanation of this is made on page 7 of the revised manuscript.

2. On page 10, give the number of academic experts who participated in the Delphi study. Currently, only the percentage is given.

The number of academics are now given on page 10.

3. It is common in Delphi studies to give panel members feedback on responses from previous rounds. Was any feedback given in this study?

Yes. Feedback was given to participants and is now mentioned on page 7.

4. Can the authors provide any quantification of agreement between academic and service user experts? The methodological article by Jorm on the Delphi method in mental health research (Aust N Z J Psychiatry. 2015 Oct;49(10):887-97) reports a number of studies that have looked at agreement among professional, consumer and carer experts. It would be informative to have such data from this study.

Thank you for this useful suggestion.

We have now measured the relationship (Pearson R correlation) between, 1) service users' and 2) academics' statement responses from Delphi round 1. The coefficient was positive and significant, but modest ($r=.25$, $p<.01$). Thus, we have added the following to the manuscript on page 11: "The two expert groups (service users and academics) showed modest agreement in their responses ($r=.25$ $p<.01$)".

We explored this further. For some statements, agreement between the two groups was high; for other statements, it was low. For example, service users deemed statements 100, 106 and 113 to be much more of a research priority than did academics. While this is interesting, agreement between two participant groups did not form part of our research question, thus we do not explore it further in our paper.

5. When reporting the results, the authors repeatedly refer to consensus across both academic and service user experts. However, the two groups were treated as one panel and only 14 out of 42 were service users. It would therefore be possible for an item to get 70% endorsement if all the academics endorsed it but only a minority of service users. I think it would only be appropriate to draw conclusions about endorsement by both groups if their consensus was assessed separately.

After conducting analysis for point 4, more reflection and re-reading the manuscript where we had mentioned agreement as per group (e.g. "both academics and service users experts.."), it was more appropriate to just say "experts agreed" as this was the case. The results section now reflects this (pages 10-12).

6. On page 12, the authors state that "The majority of experts, both academic and service users agreed...". If the consensus criterion has been set at 70%, the authors should stick to this and not use a majority (i.e. >50%) as a criterion.

We have now removed "the majority" from text on page 12 and have stuck with 70% as the consensus. The text now reads: "Experts agreed that studies looking at suicide prevention interventions in large samples, marginal groups and outpatients were research priorities in the patient safety in mental health field (83.3%, 81.0% and 81.0% respectively)."

7. I think a major limitation, which the authors don't mention, is that experts may not know what research has already been done. They were not provided with any information about the current state of research and may be suggesting areas as a priority which are already well covered. In particular, service users may not know what the existing research is, and even academic experts may not be fully aware of what is happening outside of their immediate area of expertise.

We have now added this limitation to the paper on page 16. The text now reads: "Fourthly, experts who took part in the survey were not asked about their awareness of existing research in the area. Therefore, experts, particularly service users, could have recommended research that had already been covered."

8. Another limitation is that the Delphi questionnaire was based on the interviews with 9 academics and 4 service users. There was no opportunity for the Delphi panelists to suggest additional areas for consideration in the questionnaire that were not in the interview content.

We have now added this limitation on page 16. The text now reads: "Furthermore, the survey did offer an opportunity to suggest additional areas for consideration that were not identified from the interviews."

9. On page 16, I was unclear what "attribution bias" was in this context.

Thank you for identifying this. We have now removed attribution bias from the text as it was unclear.

10. The Supplementary File gives all items, including both those that reached consensus and those that did not. I would find it easier to read the findings if those that reached consensus were presented separately from those that did not.

We have now separated out priorities reaching consensus and those who did not. They are added as supplementary file 1 and 2.

Reviewer 2

1. I thought this was a clear and interesting paper and will be helpful to take forward debate and further research in this area for example with a specific focus on community provision and particular groups e.g. children and young people. My only minor comment is that it would be helpful to clarify the key finding in the abstract which is currently so truncated as to be a bit unclear as to what the consensus actually was re top research priorities.

Thank you for highlighting this. We have now added the following to the abstract on page 2: "... however, the patient perspective on their mental healthcare is a priority."

We hope the revised version is now suitable for publication and look forward to hearing from you in due course.

Yours Sincerely,
Dr Lindsay H. Dewa

Research Associate
Imperial College London

VERSION 2 – REVIEW

REVIEWER	05-Feb-2018 University of Melbourne Australia
REVIEW RETURNED	05-Feb-2018
GENERAL COMMENTS	I think there is a missing "not" in the following sentence: "Furthermore, the survey did offer an opportunity to suggest additional areas for consideration that were not identified from the interviews." Otherwise it does not make sense as a limitation.